# A Study of Wave-Induced Effects on Sea Surface Temperature Simulations during Typhoon Events



**Zhanfeng Sun [1], Weizeng Shao [1,2],\*, Wupeng Yu [3] and Jun Li [4]**

1    College of Marine Sciences, Shanghai Ocean University, Shanghai 201306, China; zfsun@shou.edu.cn
2    National Satellite Ocean Application Service, Ministry of Natural Resources, Beijing 100081, China
3    Marine Science and Technology College, Zhejiang Ocean University, Zhoushan 316022, China; y_wupeng@outlook.com
4    East China Sea Environment Monitoring, Ministry of Natural Resources, Shanghai 201306, China; Lij@ecs.mnr.gov.cn
\*    Correspondence: shaoweizeng@mail.tsinghua.edu.cn; Tel.: +86-21-6190-0326

**Abstract:** In this work, we investigate sea surface temperature (SST) cooling under binary typhoon conditions. We particularly focus on parallel- and cross-type typhoon paths during four typhoon events: Tembin and Bolaven in 2012, and Typhoon Chan-hom and Linfa in 2015. Wave-induced effects were simulated using a third-generation numeric model, WAVEWATCH III (WW3), and were subsequently included in SST simulations using the Stony Brook Parallel Ocean Model (sbPOM). Four wave-induced effects were analyzed: breaking waves, nonbreaking waves, radiation stress, and Stokes drift. Comparison of WW3-simulated significant wave height (SWH) data with measurements from the Jason-2 altimeter showed that the root mean square error (RMSE) was less than 0.6 m with a correlation (COR) of 0.9. When the four typhoon-wave-induced effects were included in sbPOM simulations, the simulated SSTs had an RMSE of 1 °C with a COR of 0.99 as compared to the Argos data. This was better than the RMSE and COR recovered between the measured and simulated SSTs, which were 1.4 °C and 0.96, respectively, when the four terms were not included. In particular, our results show that the effects of Stokes drift, as well as of nonbreaking waves, were an important factor in SST reduction during binary typhoons. The horizontal profile of the sbPOM-simulated SST for parallel-type typhoon paths (Typhoons Tembin and Bolaven) suggested that the observed finger pattern of SST cooling (up to 2 °C) was probably caused by drag from typhoon Tembin. SST was reduced by up to 4 °C for cross-type typhoon paths (Typhoons Chan-hom and Linfa). In general, mixing significantly increased when the four wave-induced effects were included. The vertical profile of SST indicated that disturbance depth increased (up to 100 m) for cross-type typhoon paths because the mixing intensity was greater for cross-type typhoons than for parallel-type typhoons.

**Keywords:** typhoon wave; sea surface temperature; WAVEWATCH-III; sbPOM

## 1. Introduction

Typhoons occur frequently in the Western Pacific Ocean (WP) [1], affecting energy exchange at the air–sea boundary layer [2,3] and leading to several secondary hazards, such as extreme waves [4,5], landslides, and heavy rains [6]. Binary typhoons, in which two storms of tropical cyclone intensity or more occur simultaneously, have also been recorded in the WP. Because of the strong wind interactions endemic to binary typhoons, binary typhoons have more complicated effects on the sea surface than single typhoons due to the influence of total heat flux exchange on the upper ocean response [7,8]. At present, moored buoys [9] and satellites [10,11] provide real-time observations of oceanic conditions, particularly winds and waves, during hurricanes and typhoons. However, those devices are unable to generate time-series data with a fine spatial resolution; that is, the resolution of a scatterometer is typically 12.5 km [12], while that of an altimeter

is 10 km [13]. Thus, the data collected by these devices are not sufficient for long-term distribution analyses.

Over recent decades, as wave theory and computation technologies have matured, several numeric wave models have been proposed. At the beginning of the 1980s, WAMDIG initially proposed the third-generation numeric wave ocean model (WAM) [14], which integrated basic wave propagation effects to describe the evolution of a two-dimensional ocean wave spectrum. Subsequent authors developed the WAVEWATCH-III (WW3) [15] and Simulating Waves Nearshore (SWAN) [16] models based on the principles of the WAM. The main difference between the WW3 and SWAN models is the applicability of the model: SWAN was originally developed as a nearshore model, while WW3 was developed for the oceanic scales. Therefore, the WW3 model is usually employed for wave simulations over large regions, such as global seas [17] or the western Pacific Ocean [18], while the SWAN model is typical used to analyze coastal waters [19]. Model-simulated waves are also commonly used in synthetic aperture radar (SAR) wave monitoring [20,21] and, in particular, as auxiliary data for typhoon analysis [22].

During typhoon events, the sea state is complicated due to strong, synoptic-scale air–sea interactions and turbulent mixing at the sea surface [23]. In particular, sea surface temperatures (SSTs) during typhoons are rarely measured using real-time techniques such as Argos [24]. This air–sea mixing, as well as the Ekman pumping induced by cyclonic vorticity, deepens the mixed layer at the sea surface, decreasing SST [25–27]. Remarkably, observational data have shown a maximum SST cooling of 9 °C [28], and short-term satellite data recorded during two typhoon events revealed anomalously cold SST patches that were up to 6 °C colder than the SSTs of the surrounding warm tropical sea [29]. SST cooling in turn increases typhoon intensity and movement by modulating energy fluxes and stability at the air–sea boundary layer [30]. Therefore, SST cooling is one of the more noticeable oceanic responses to a moving typhoon due to its significant influence on oceanic and atmospheric dynamics [31]. Due to the limited availability of observational data during typhoons, the coupled atmospheric–oceanic model provides a powerful alternative way to study SST cooling and its impact on typhoon intensity [32].

Recent numerical experiments have aimed to clarify the unique characteristics and underlying mechanisms of SST cooling [33,34]. Two case studies [35,36] have suggested that SST cooling in the inner-core region of the cyclone, which is defined as within a 111-km radius of the cyclone center [37], may weaken typhoon intensity. However, the largest SST reduction often occurs in the right-rear quadrant of the typhoon. Although sea-surface waves themselves act over small scales, ranging from meters to kilometers, wave-induced effects, such as breaking waves, nonbreaking waves, radiation stress, and Stokes drift, affect the air–sea energy exchange at the boundary layer, especially during strong winds. Thus, wave-induced effects should be considered in analyses of SST cooling. The produced cooling is a function of both typhoon forward speed and intensity. Generally, the lower the forward speed, the higher the cooling rate and the higher the intensity, meaning a larger cooling rate is expected [38,39]. The extra cooling and turbulent mixing on the right side of the track in the Northern Hemisphere as a result of the rightward bias can contribute to a larger deepening of the mixed layer [40]. In most cases, binary typhoons are stronger than single typhoons in terms of duration and range, and will cause strong upwelling and mesoscale cyclone vortices in certain areas, which will also have a greater impact on SST [41]. Furthermore, it is important to assess SST cooling during binary typhoons, which include both parallel- and cross-type movements.

The ocean circulation model, which has been termed the Princeton Ocean Model (POM), is commonly used to simulate global marine dynamics [42,43], such as current and SST. An updated version of POM, the Stony Brook Parallel Ocean Model (sbPOM) [44], has been improved and enhanced using the parallel computation technique. The scalability of the POM model is better than that of its predecessors. In this study, we simulated wave fields during certain typhoon events using the WW3 model and calculated four of the effects of strong winds: breaking waves, nonbreaking waves, radiation stress, and

Stokes drift. Subsequently, SST was simulated using an sbPOM that included these four factors. In particular, we focused on fluctuations in SST cooling during various types of typhoon movement.

Indeed, the primary aim of this study was to assess SST cooling during binary typhoons with different paths. The datasets used, which are described in Section 2, include the typhoon events, the forcing wind fields, the open boundary conditions for modeling, and measurements from the Jason-2 altimeter and Argos. The model settings for the WW3 and sbPOM simulations are also given in Section 2. Waves were simulated using the WW3 model and SSTs were simulated using sbPOM, both based on the four effects of the typhoon waves. We assessed the accuracies of these models and the discussions in Section 3. Our conclusions are summarized in Section 4.

## 2. Materials and Methods

Four typhoons that passed through the China Sea were analyzed in this study: Tembin, Bolaven, Chan-hom, and Linfa. The tracks of these typhoons were identified as either parallel- or cross-type. Superimposition of the tracks of these four typhoons over the water depths obtained from the General Bathymetry Chart of the Oceans (GEBCO) data showed that the paths of Tembin and Bolaven (19–30 August 2012) were almost parallel, while the path of Chan-hom intersected that of Linfa (7–12 July 2015; Figure 1).

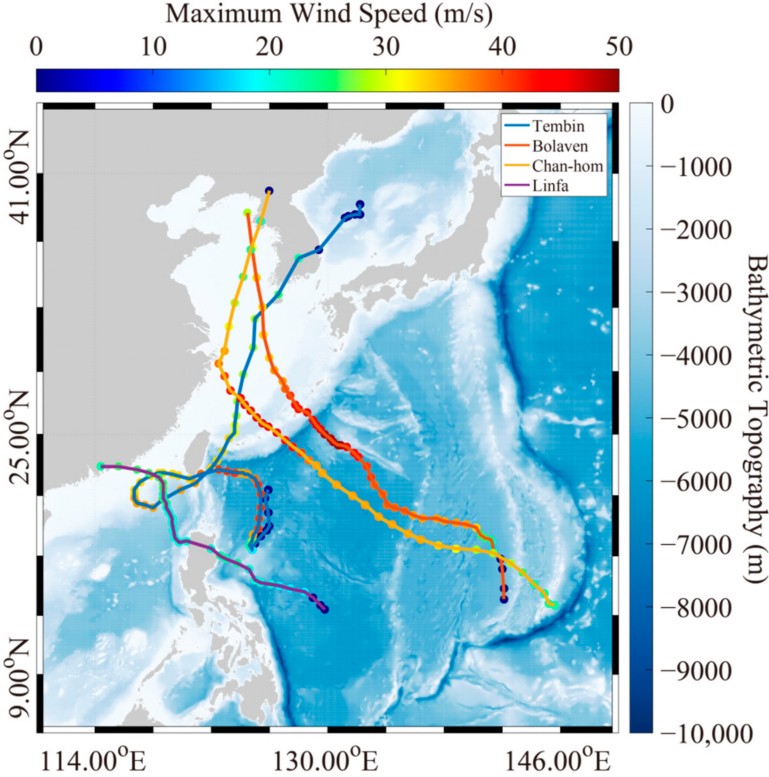

**Figure 1.** The tracks of the typhoons analyzed in this study superimposed on the water depth data from the General Bathymetry Chart of the Oceans (GEBCO).

Two numeric models, a WW3 model and a sbPOM, were implemented for the China Seas. The European Centre for Medium-Range Weather Forecast (ECMWF) has provided operational products for the global seas since 1958; ECMWF data have a high resolution (up to 0.125°) at intervals of 1 h. Using the ECMWF wind data, we hindcasted the long-term wave distributions previously simulated using SWAN and WW3 models in two Chinese seas: the Bobai Sea [45] and the South China Sea [46]. The results indicated that the model-simulated waves were consistent with buoy data and altimeter measurements. However, simulated measurements were systematically underestimated as compared to observational

measurements [47]. In a recent study, we reconstructed "H-E winds," composited of ECMWF winds and a parametric Holland model [15], for typhoons. The model was trained by fitting the shape parameter to buoy-measured observations. We then compared simulated wind speeds with those measured by moored buoys, and found that the root mean square error (RMSE) of wind speed was less than 3 m/s for the shape parameter equivalent to 0.4 [15]. A representative example of H-E wind fields, for Typhoons Chan-hom and Linfa at 18:00 UTC on 5 July 2015, is shown in Figure 2. In this figure, two typhoon centers are clearly apparent in the H-E wind fields. Importantly, the underestimation of ECMWF winds was improved by reanalysis using H-E winds.

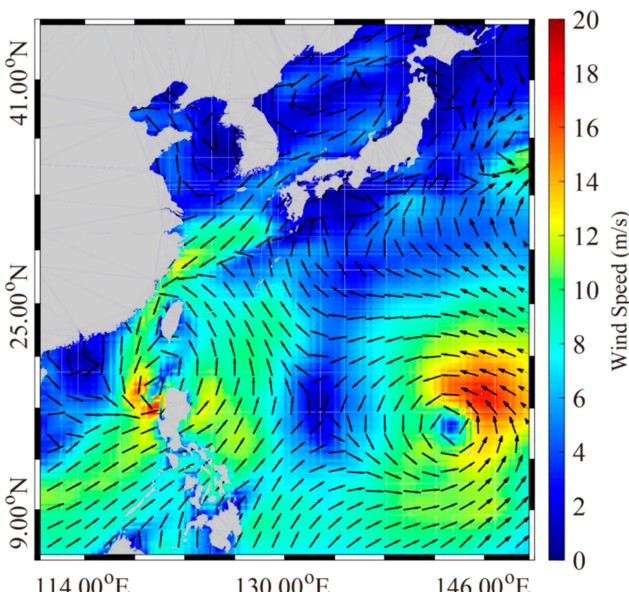

**Figure 2.** The wind map for Typhoons Chan-hom and Linfa at 18:00 UTC on 5 July 2015. Wind maps were composited using the European Centre for Medium-Range Weather Forecasts (ECMWF) wind data and a parametric Holland model (H-E).

The initial fields for the SST simulation of sbPOM were the monthly average SST and the sea surface salinity from the Simple Ocean Data Assimilation (SODA) data, which have a spatial grid resolution of 0.5°. A representative example from August 2012 is shown in Figure 3. The upper boundary forcing fields were obtained based on the National Centers for Environmental Prediction (NCEP) total heat flux parameters (latent heat flux, sensible heat flux, long-wave radiation, and short-wave radiation) at 6 h intervals and a spatial resolution of 1.875° × 1.905° (longitude × latitude); in contrast, ECMWP provides flux data twice per day. Argos is an international cooperative project begun in 2000. The Argos project, which aims to profile ocean temperature and salinity, is a major component of many ocean observation systems [48]. In this study, we used the high-quality SST measurements from Argo project to validate the simulations of the sbPOM. As an example, the map of NCEP total heat flux at 18:00 UTC on 5 July 2015 is shown in Figure 4; in this figure, triangles represent the geographic locations of the available Argos stations (>20) used in this study.

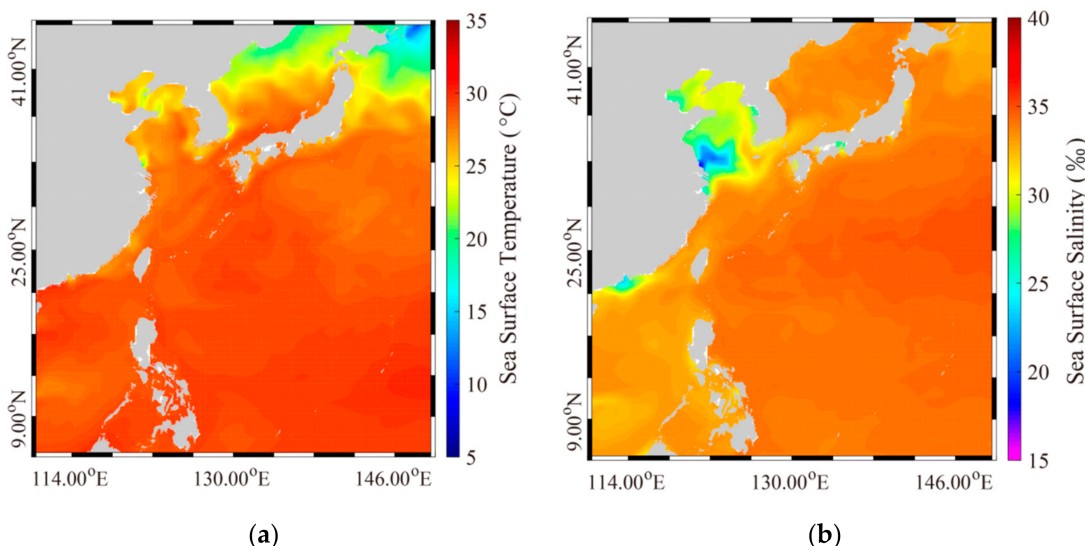

**Figure 3.** The Simple Ocean Data Assimilation (SODA) data used in the Stony Brook Parallel Ocean Model (sbPOM) on August 2012. (**a**) Monthly average sea surface temperature. (**b**) Monthly average sea surface salinity.

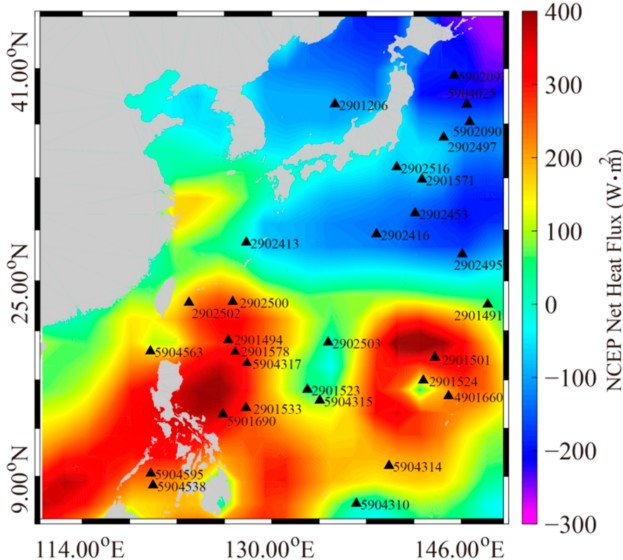

**Figure 4.** The National Centers for Environmental Prediction (NCEP) total heat flux at 18:00 UTC on 5 July 2015, overlaid with the geographic locations of the Argos stations used in this study.

In addition, SWH measurements were obtained from the altimeter Jason-2, which is a follow-on satellite to that of the Jason-1 oceanography mission of the National Aeronautics and Space Administration (NASA) and is a valuable source of global wave distributions [49]. These SWH data were used to validate the WW3-simulated SWH data. The WW3-simulated SWH map was overlaid the data from the Jason-2 altimeter at 18:00 UTC on 5 July 2015 (Figure 5). In this figure, the two cyclone-induced wave patterns are clearly visible. The basic settings of the WW3 model and the sbPOM (e.g., forcing fields, open boundary conditions, and output resolution) are summarized in Table 1. The details of the WW3 model and the sbPOM are given in Appendices A and B, respectively.

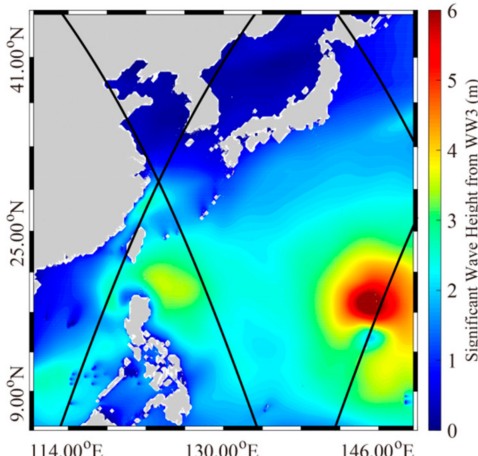

**Figure 5.** The significant wave height (SWH) data obtained from the WAVEWATCH-III (WW3) model at 18:00 UTC on 5 July 2015, overlaid with the footprints of the Jason-2 altimeter.

**Table 1.** The basic settings for the WAVEWATCH-III (WW3) model and the Stony Brook Parallel Ocean Model (sbPOM).

| | Forcing Fields | Output Resolution | Open Boundary Conditions |
|---|---|---|---|
| WW3 | Winds composited using the European Centre for Medium-Range Weather Forecast (ECMWF) data and a parametric Holland model (H-E) | Temporal resolution of 30 min and spatial grid resolution of 0.1° | / |
| sbPOM | H-E winds; Simple Ocean Data Assimilation (SODA) sea surface temperature and salinity Wave-induced: breaking wave; nonbreaking wave Radiation stress; stokes drift | Temporal resolution of 30 min and spatial grid resolution of 0.25° | National Centers for Environmental Prediction (NCEP) latent heat flux |
| | | | NCEP sensible heat flux |
| | | | NCEP long-wave radiation |
| | | | NCEP short-wave radiation |

## 3. Results and Discussions

### 3.1. Validation of the SWHs Simulated Using the WW3 Model

The wave fields during the four typhoon events were simulated using WW3 model. The WW3-simulated SWHs were compared with the SWH measurements of the Jason-2 altimeter statistically using root mean squared error (RMSE) and correlations (CORs). RMSE and COR were calculated as follows:

$$\text{RMSE} = \sqrt{\frac{\sum_{i-1}^{N}\left(X_{mod}^{i} - X_{obs}^{i}\right)^2}{N}} \text{ and} \tag{1}$$

$$\text{COR} = \frac{\text{Cov}(X_{mod}, X_{obs})}{\sqrt{\text{Var}(X_{mod})\text{Var}(X_{obs})}}, \tag{2}$$

where N was the number of matchups, $X_{mod}$ were the model-simulated results, $X_{obs}$ were the measurements from the Jason-2 altimeter, Cov was the covariance, and Var was the variance.

The analysis of more than 10,000 matchups during Typhoons Tembin and Bolaven (19–30 August 2012) resulted in SWHs with an RMSE of 0.54 and a COR of 0.91 (Figure 6a). Similarly, the SWHs of Typhoons Chan-hom and Linfa (2–11 July 2015) had an RMSE of 0.50 m and a COR of 0.90 (Figure 6b). These results indicate that the WW3-simulated wave fields during typhoons were reliable, even under extreme conditions (SWH > 8). The methods used to calculate breaking waves, nonbreaking waves, radiation stress, and Stokes

drift are given in Appendix A using WW3-simulated parameters (e.g., SWH, mean wave period, wavelength, and dominant wave propagation velocity).

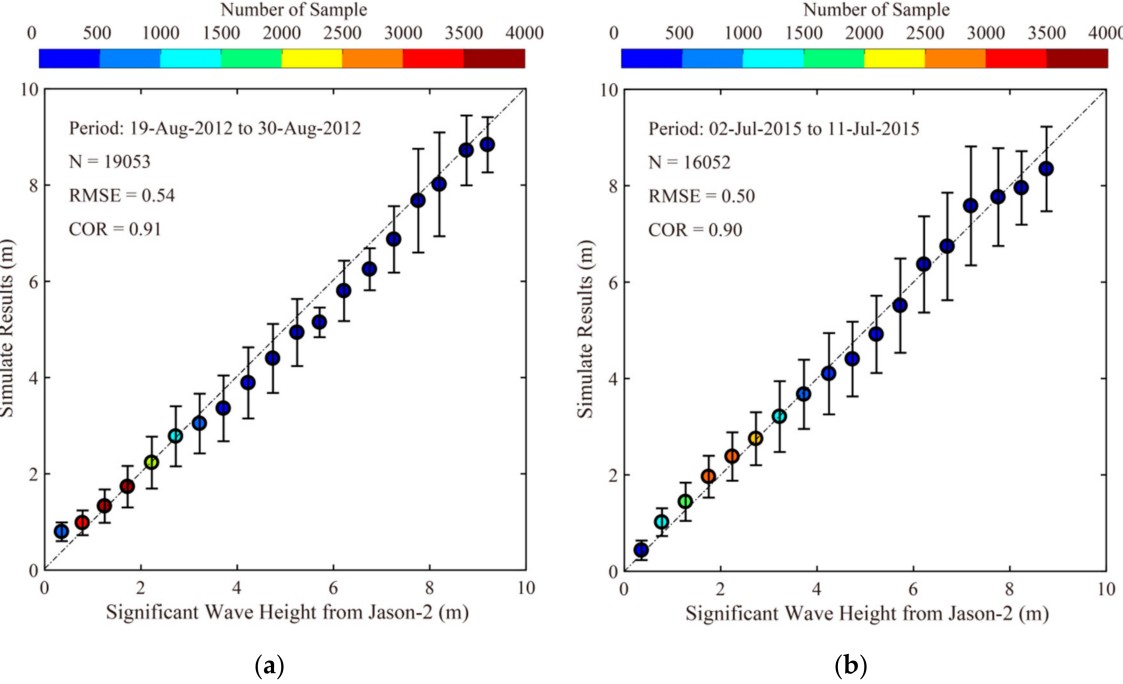

(**a**)　　　　　　　　　　　　　　(**b**)

**Figure 6.** Comparisons of WW3-simulated SWHs with measurements from the Jason-2 altimeter in 0.5-m bins between 0 and 10 m. (**a**) 19–30 August 2012; (**b**) 2–11 July 2015.

### 3.2. Analysis of SSTs Simulated Using the sbPOM

The four effects, which were empirically calculated based on wave parameters derived from the WW3 model, were treated as forcing fields in the sbPOM. Here, we first present the results of the sbPOM, including the individual wave-induced terms simulated using the WW3 model. When the simulated results were compared with the Argos measurements during the four typhoons, the bias in nonbreaking stress and Strokes drift terms was about 1 °C; the bias in the breaking and radiation stress terms was greater (Table 2).

**Table 2.** Statistical comparisons of sea surface temperatures (SSTs) generated by the sbPOM and Argos during the four typhoons.

|  | Wave Breaking | Nonbreaking Wave | Radiation Stress | Stokes Drift |
|---|---|---|---|---|
| RMSE (°C) | 1.23 | 1.16 | 1.40 | 1.02 |
| COR | 0.96 | 0.97 | 0.97 | 0.98 |

We included Stokes drift, accounting for depth decay, in the sbPOM. The interaction between friction velocity and Stokes drift may result in Langmuir circulations in binary typhoons, and may subsequently affect SST. This is probably why Stokes drift was the least biased term.

We comparted the SSTs from the available Argos data with those simulated using the sbPOM in 2 °C bins between 0 and 30 °C during the Tembin and Bolaven typhoons (19–30 August 2012; Figure 7). Figure 7a shows SSTs simulated without the four effects induced by typhoon waves (RMSE of 1.42 °C and COR of 0.95), while Figure 7b shows SSTs simulated including the four effects induced by typhoon waves (RMSE of 1.07 °C and COR of 0.99). Statistical analyses of the Chan-hom and Linfa typhoons (2–11 July 2015) yielded similar results: the results of the analysis including the four effects induced by typhoon waves (RMSE of 1.01 °C and COR of 0.99; Figure 8a) were better than those not

including the four effects induced by typhoon waves (RSME of 1.37 °C and COR of 0.96; Figure 8b). Therefore, we concluded that the accuracy of the sbPOM SST simulation was improved when the four WW3-simulated effects induced by typhoon waves were included. The RMSE of SST was 1.40 °C when radiation stress was included, which was similar to the results generated without including any wave-induced effects. This suggested that radiation stress had little influence on SST cooling. Nonbreaking wave-induced mixing alone led to SST cooling in a study of individual typhoons [50]. However, our results show that effects of Stokes drift were also an important factor in SST cooling during binary typhoons.

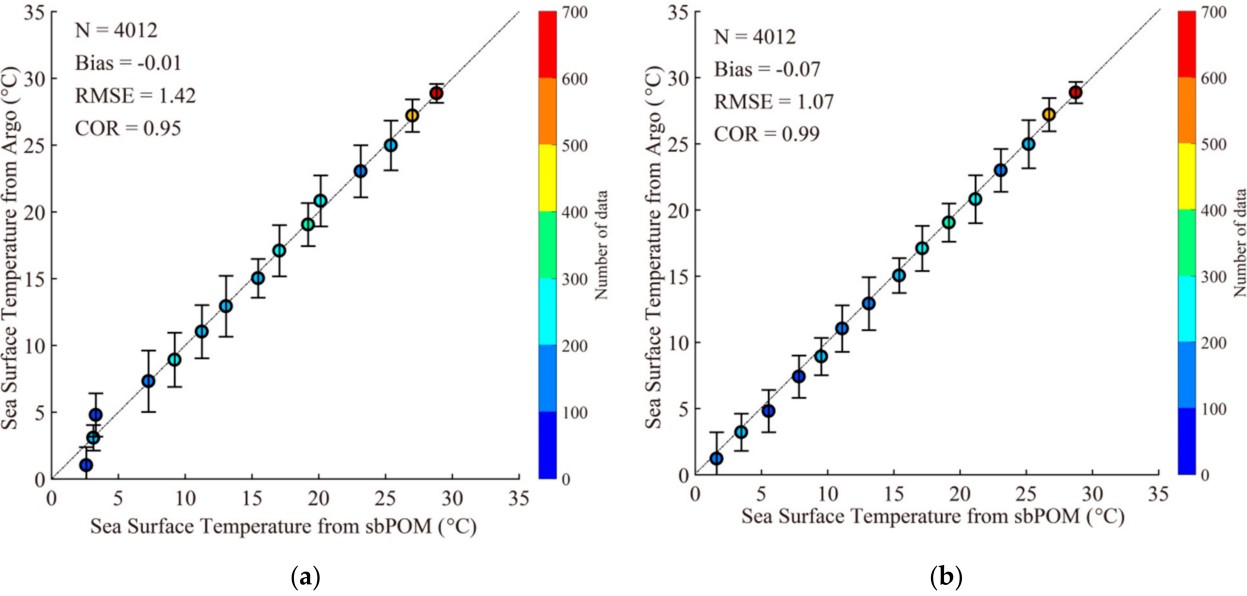

**Figure 7.** Comparisons of the SSTs from Argos with SSTs simulated using the sbPOM for 2 °C bins between 0 and 30 °C during the period 19–30 August 2012. (**a**) Not including the four effects induced by typhoon waves. (**b**) Including the four effects induced by typhoon waves.

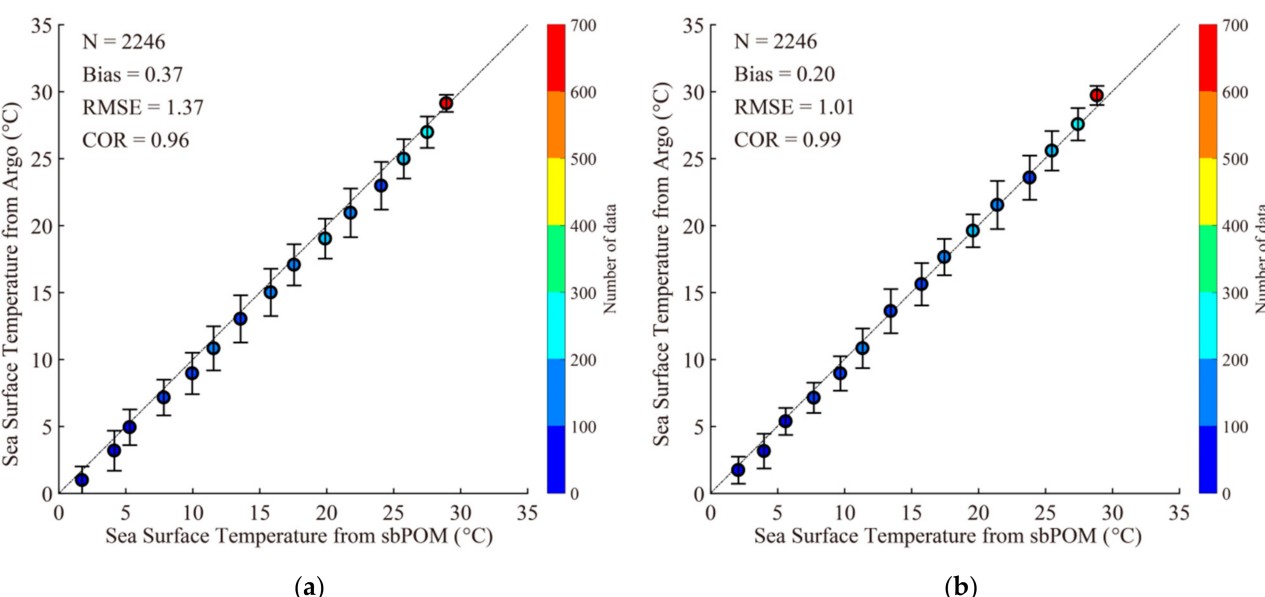

**Figure 8.** Comparisons of the SSTs from Argos with SSTs simulated using the sbPOM for 2 °C bins between 0 and 30 °C during the period 2–11 July 2015. (**a**) Not including the four effects induced by typhoon waves. (**b**) Including the four effects induced by typhoon waves.

*3.3. Discussions*

Several cases of typhoon-associated SST cooling have been analyzed using numeric modeling [51] and satellite observations [52]. These previous studies have shown that the paths of binary typhoons can be divided into six categories, which are also related to the intensity of the airflow [53]. SST cooling occurs to the right of the typhoon path in the Northern Hemisphere, and the return of SST to normal depends on the thickness of the upper ocean layer and the wind conditions [54]. These findings suggest that SST cooling patterns will be more complicated during binary typhoons, especially simultaneous typhoon events, due to the interactions between the two wind and wave systems.

The daily average SST distributions during the Tembin and Bolaven typhoons (red lines) on 26–30 August 2012 are shown in Figure 9a–d. Similarly, Figure 10 shows the daily average SST distributions during the Chan-hom and Linfa typhoons on 7–11 July 2015. The reduction in SST associated with cross-type typhoon paths (Figure 9b,c) was up to 4 °C, while the reduction in SST associated with parallel-type typhoon paths was up to 2 °C. The finger pattern of SST cooling shown in Figure 9a was probably caused by drag from Typhoon Tembin. The mixing associated with the selected typhoons, including four effects, is shown at Site A (Figure 11a) and at Site B (Figure 11b). In these figures, $K_h$ represents the mixing induced by heat flux and $K_m$ represents the mixing induced by momentum. Generally, mixing significantly increased during parallel-type typhoons when the four wave-induced effects were included. In particular, mixing intensity up to a depth of 50 m was greater for cross-type typhoons (maximum 0.2) than for parallel-type typhoons (maximum 0.1). This suggested that the strong energy exchange associated with cross-type typhoon paths led to the substantial reduction in SST, and that $K_h$ was a major factor affecting the mixing associated with both types of typhoon paths.

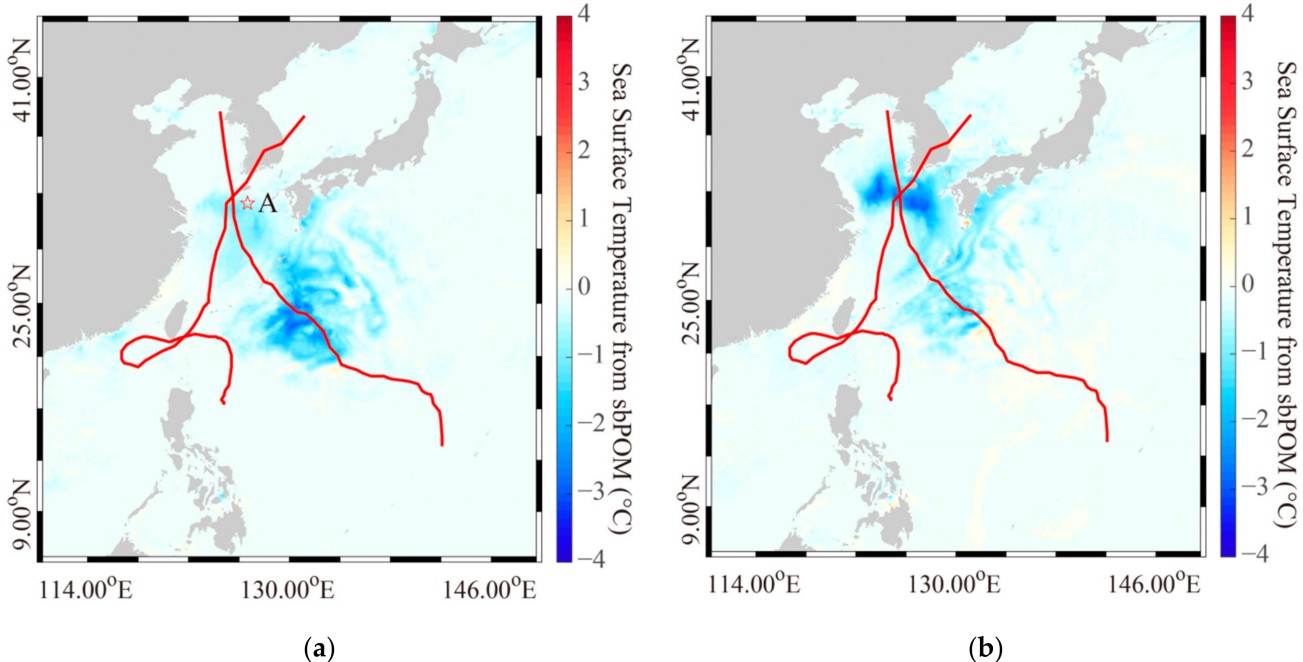

(**a**)                                                                 (**b**)

**Figure 9.** *Cont.*

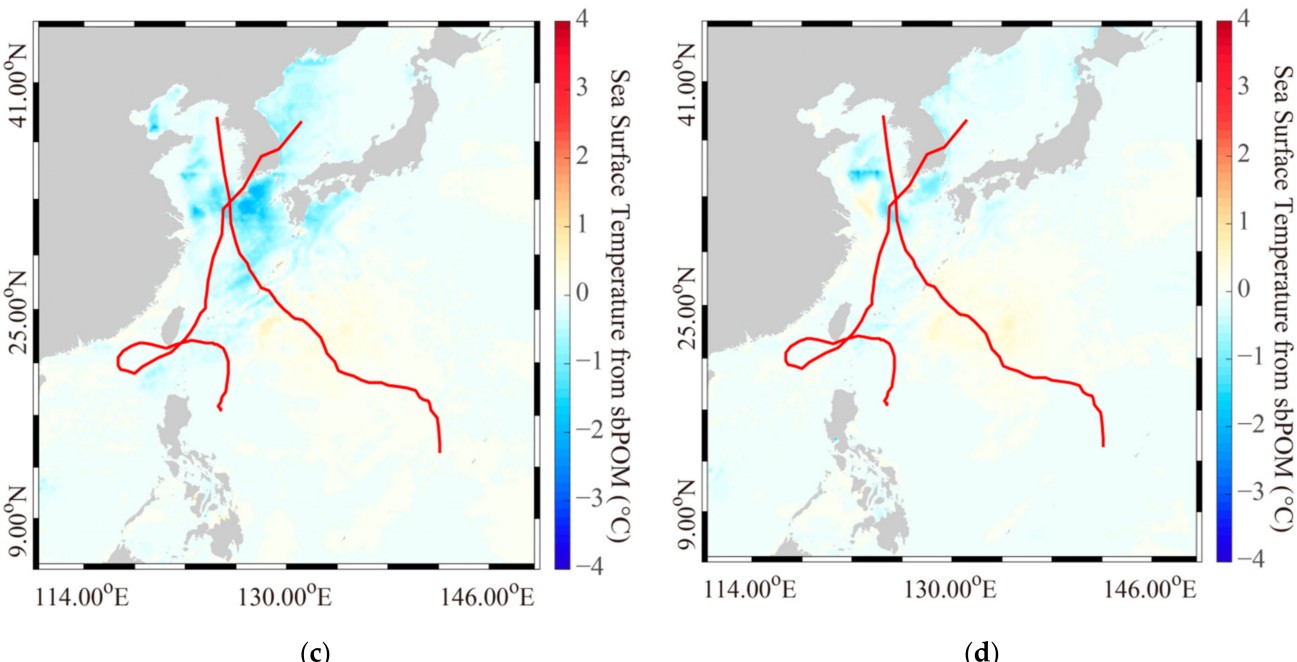

**Figure 9.** Daily average SSTs simulated using the sbPOM during Typhoons Tembin and Bolaven (26–30 August 2012). (**a**) Results from 26 July minus results from 27 July; (**b**) results from 27 July minus results from 28 July; (**c**) results from 28 July minus results from 29 July; (**d**) results from 29 July minus results from 30 July.

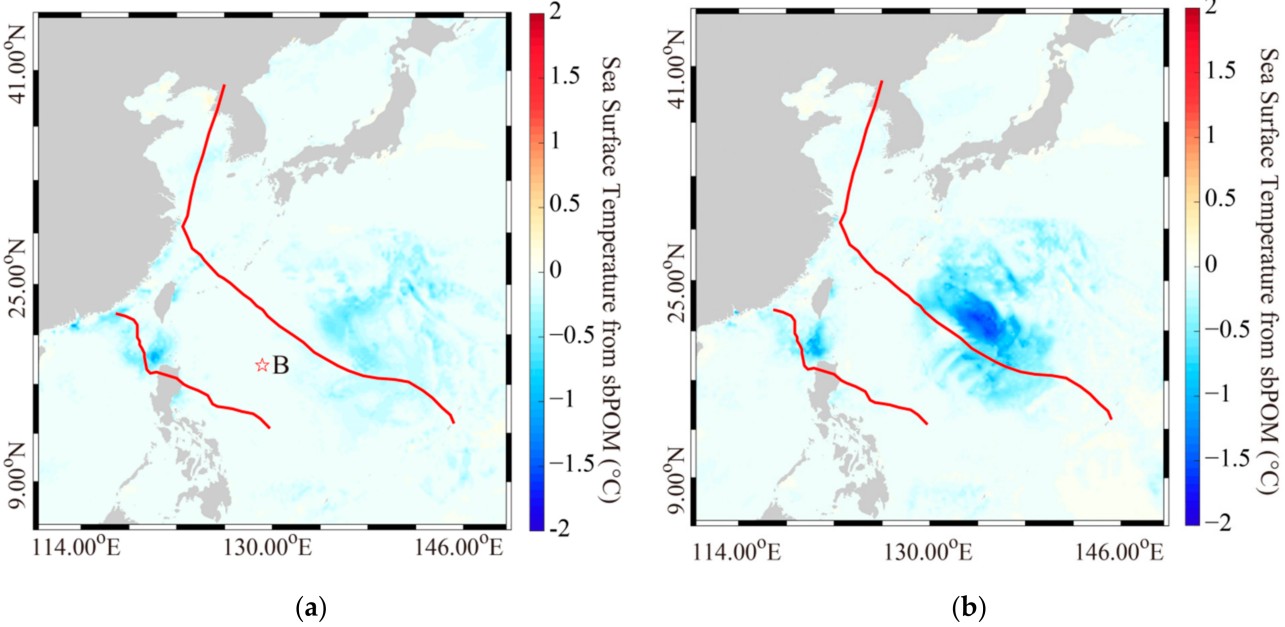

**Figure 10.** *Cont.*

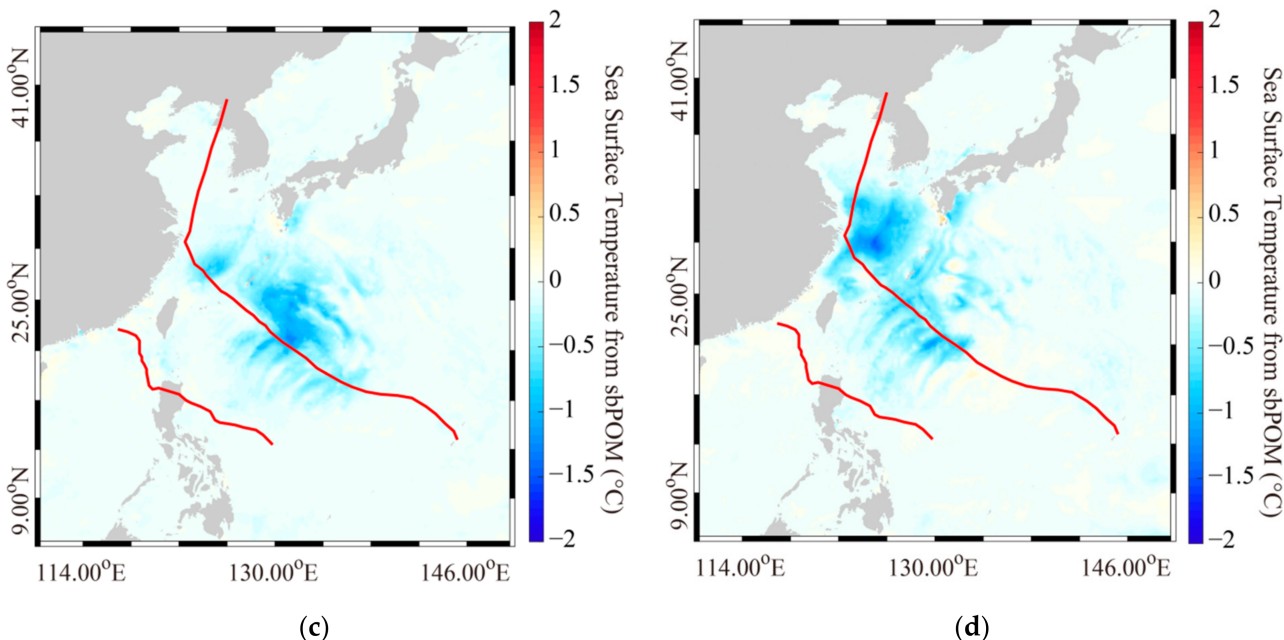

(c)                                                              (d)

**Figure 10.** Daily average SSTs simulated using the sbPOM during the Chan-hom and Linfa typhoons on 7–11 July 2015. (**a**) Results from 7 July minus results from 8 July; (**b**) results from 8 July minus results from 9 July; (**c**) results from 9 July minus results from 10 July; (**d**) results from 10 July minus results from 11 July.

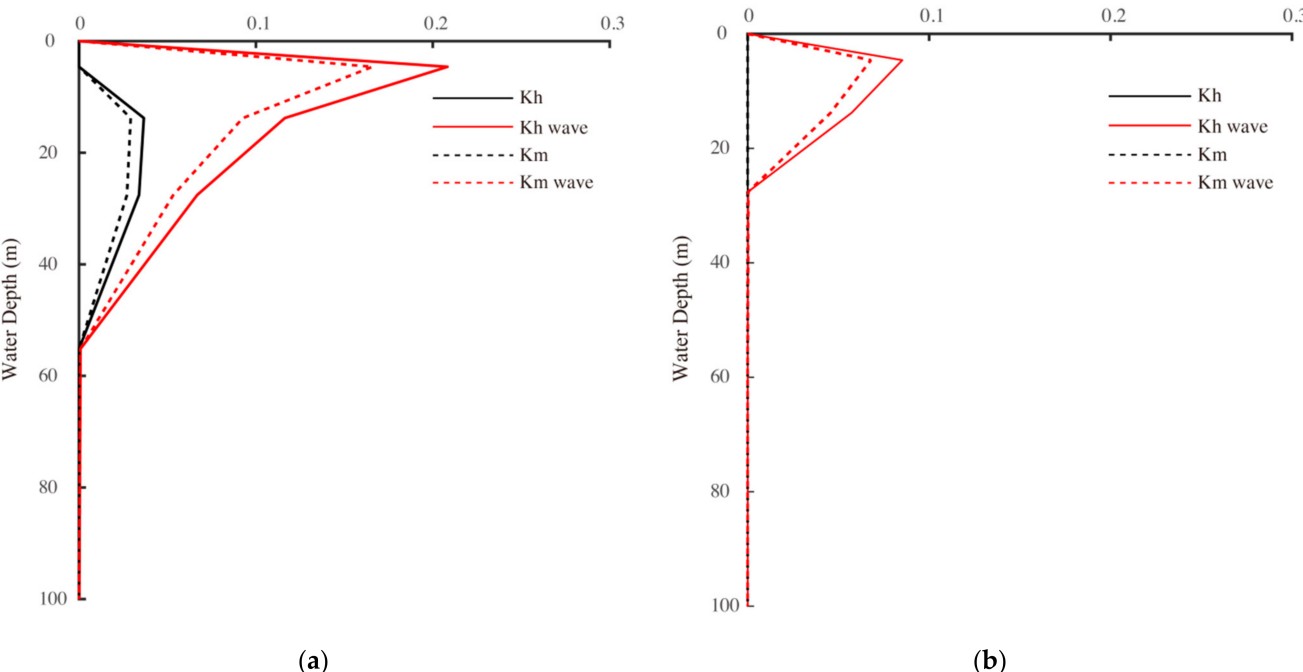

(a)                                                              (b)

**Figure 11.** (**a**) The vertical mixing profile on 27 and 28 August 2012 at Site A in Figure 10a, and (**b**) the vertical mixing profile on 10 and 11 July 2015 at Site B in (**a**). Note that $K_h$ represents the mixing induced by heat flux, and $K_m$ represents the mixing induced by momentum.

Two sites (A and B) were selected to further investigate the vertical profile of temperature cooling (Figure 12). At Site B (Figure 12b), which is near the point where the two typhoon paths crossed, the depth of the typhoon-induced disturbance reached 150 m. The vertical temperature profile simulated by sbPOM for the parallel-type typhoon path indicated that the maximum depth of the typhoon-induced disturbance was about 50 m at

Site A (Figure 12a), and that the wave-induced effects of water temperature weakened with increasing depth. Thus, temperature reduction and typhoon-induced disturbance depth were both greater for cross-type typhoon paths than for parallel-type typhoon paths. A previous one-dimensional ocean mixed layer model (OMLM-Noh) [55] showed that the mixed layer deepened to 45 and 25 m for the typhoons Hagibis and Mitag, respectively. Our work may have identified a deeper mixed layer because the one-dimensional mixed layer model only considers mixing effects, whereas our model included Stokes drift effects.

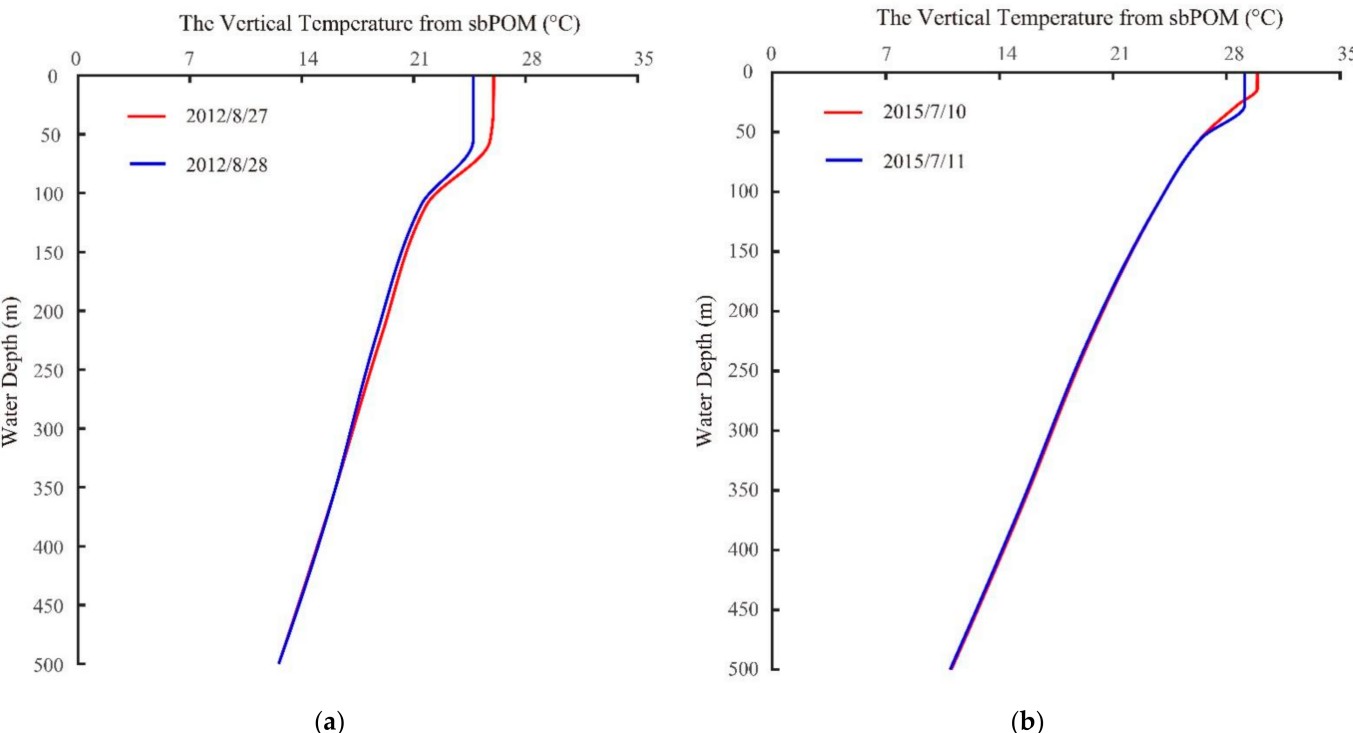

**Figure 12.** (**a**) The vertical temperature profile simulated using the sbPOM on 27 and 28 August 2012 at Site A in Figure 10a, and (**b**) the vertical temperature profile simulated using the sbPOM on 10 and 11 July 2015 at Site B in Figure 11a.

## 4. Summary and Conclusions

The aim of this work was to investigate SST cooling in response to different typhoon paths (i.e., parallel-type and cross-type) using two marine models: a wave model (WW3) and a circulation model (sbPOM). Previous studies [7,8,56] have considered the effects of typhoon waves, such as nonbreaking waves and radiation stress, on SST cooling individually. Here, we simulated SST using an sbPOM that included four effects of typhoon waves simulated by the WW3 model (breaking waves, nonbreaking waves, radiation stress, and Stokes drift), which were stronger than those at regular sea sites. We also investigated the horizontal and vertical distributions of SST cooling.

Composite H-E winds, which combine cyclonic winds using parametric Holland and ECMWF reanalysis data, were treated as the forcing field in the WW3 model. The WW3-simulated SWHs were validated against the measurements of Jason-2 altimeter, and an RMSE of less than 0.6 m and a COR of about 0.9, indicating that WW3-simulated waves were suitable for use in this study. The typhoon-wave-induced effects were calculated based on several parameters, such as SWH, mean wave period, wavelength, and dominant wave propagation velocity. When the four effects induced by typhoon waves were considered, the bias in the SST results obtained via the sbPOM simulation was improved by about 0.5 °C as compared to simulations based on Argos measurements. As noted in a previous study [46], nonbreaking waves reduce SST for individual typhoons. However, the effects of Stokes drift also have an important influence on SST cooling during binary typhoons.

Therefore, we concluded that typhoon-wave-induced effects should be included in SST simulations during typhoons, because typhoon waves influence the air–sea boundary layer (via breaking waves and Strokes drift), as well as the mixing layer (via nonbreaking waves and radiation stress).

The daily average SST distributions during the four typhoons were analyzed. We identified a finger pattern of SST cooling during both parallel-type and cross-type typhoons. SST was reduced up to 2 °C for parallel-type typhoons and up to 4 °C for cross-type typhoons. Mixing was significantly enhanced when wave-induced effects were considered; the mixing induced by heat flux was stronger than that induced by momentum. In addition, the mixing associated with cross-type typhoons was greater than that associated with parallel-type typhoons. Vertical SST profiles during the four typhoons were also studied. The results suggest that the typhoon-induced disturbance depth was 100 m for cross-type typhoons, which was deeper than the disturbance depth associated with parallel-type typhoons (50 m).

In future studies, we aim to consider sea-surface roughness and the air–sea energy exchange, including variations in the drag coefficient, using the same numeric models (WW3 and sbPOM) under binary typhoon conditions.

**Author Contributions:** Conceptualization, W.S. and Z.S.; methodology, W.S., Z.S. and W.Y.; validation, Z.S., W.Y. and J.L.; formal analysis, W.S. and Z.S.; investigation, Z.S.; resources, W.S.; writing—original draft preparation, Z.S. and W.S.; writing—review and editing, W.S.; visualization, W.Y. and J.L.; funding acquisition, W.S. All authors have read and agreed to the published version of the manuscript.

**Funding:** This research was funded by the National Key Research and Development Program of China under contract nos. 2017YFA0604901 and 2017YFA0604904, the National Natural Science Foundation of China under contract nos. 41806005 and 42076238, the National Social Science Foundation of China (Major Program) contract no. 15ZDB17 and the Science and Technology Project of Zhoushan City, China, under contract no. 2019C21008.

**Institutional Review Board Statement:** Not applicable.

**Informed Consent Statement:** Not applicable.

**Data Availability Statement:** Due to the nature of this research, participants of this study did not agree to share the data publicly, so supporting data are not available.

**Acknowledgments:** We are truly thankful for the National Centers for Environmental Prediction (NCEP) of the National Oceanic and Atmospheric Administration (NOAA) providing the source for the WAVEWATCH-III (WW3) model. The original code of Stony Brook Parallel Ocean Model (sbPOM) is available via http://www.ccpo.odu.edu (accessed on 3 June 2021) The European Centre for Medium-Range Weather Forecasts (ECMWF) provides wind data via http://www.ecmwf.int (accessed on 3 June 2021). General Bathymetry Chart of the Oceans (GEBCO) data are downloaded via ftp.edcftp.cr.usgs.gov. The Simple Ocean Data Assimilation (SODA) data are collected via https://climatedataguide.ucar.edu (accessed on 3 June 2021). The NCEP wind field and heat flux is collected via http://www.cdc.noaa.gov (accessed on 3 June 2021). The measurements from altimeter Jason-2 and Argos are accessed via https://data.nodc.noaa.gov (accessed on 3 June 2021) and http://www.argodatamgt.org (accessed on 3 June 2021), respectively.

**Conflicts of Interest:** The authors declare no conflict of interest.

## Appendix A

The basic wave propagation balance equations in the WAVEWATCH-III (WW3) model can be briefly described as follows:

$$\frac{D}{Dt}(N(k, \theta; x, t)) = \frac{S(k, \theta; x, t)}{\sigma} \text{ and} \tag{A1}$$

$$S = S_{in} + S_{nl} + S_{ds} + S_{bot} + S_{db}, \tag{A2}$$

where the wavenumber-direction spectrum N is the basic spectrum of WW3 in terms of wavenumber k, wave propagation direction θ, space dimension x, and time dimension t; N is the wave action density spectrum; σ is the intrinsic frequency; and S(k,θ;x,t) describes the net of sources and sinks from the wavenumber-direction spectrum. The sink term S(k,θ;x,t) contains the impacts of linear and nonlinear wave propagation energy, including wind energy input $S_{in}$, a wave–wave interaction term $S_{nl}$, dissipation $S_{ds}$, and the empirical parameterizations of wave–bottom friction $S_{bot}$ and depth-induced breaking $S_{db}$. The parameterizations of these terms are conveniently provided by the WW3 model, as described in the technical manual. The four wave-induced effects were calculated based on several parameters. Specifically, the parametrization of input/dissipation terms is referred to switch ST2+STAB2 [13,57], and the packages for processing nonlinear terms on quadruple wave–wave interactions is referred to switch DIA [5,58].

**Theoretical expression of a breaking wave:**

The energy dissipation rate due to wave breaking in a unit area of water column can be expressed as

$$R_{ds} = \rho_w g \int S_{ds}(\mathbf{k}) d\mathbf{k}, \tag{A3}$$

where $\rho_w$ is the density of sea water, g is the gravity constant (9.8 m/s$^2$), and $R_{ds}$ is the downward input of turbulent kinetic energy flux due to wave breaking on the sea surface. In practice, a unified analytical form of wave breaking energy dissipation is derived as follows [59]:

$$R_{dis} = 2.97\gamma\rho_w g\beta_*^{-2}\omega_p E, \tag{A4}$$

where

$$\beta_* = \frac{g}{u_*\omega_p}, \tag{A5}$$

$$E = \int F(\mathbf{k}) d\mathbf{k}, \tag{A6}$$

$$\omega_p = \frac{2\pi}{T_p}, \tag{A7}$$

where $\gamma$ is energy dissipated per unit of white crown accounting for the percentage of total wave energy dissipation, generally $\gamma = 0.1$; $\beta_*$ is the wave age; E is spectral density; $\mathbf{k}$ is wave number vector; $F(\mathbf{k})$ is the two-dimensional wave spectrum; $\omega_p$ is the spectral peak angular frequency; $u_*$ is the friction velocity $(=\sqrt{c_d U_{10}})$; $c_d$ is the drag coefficient; and $T_p$ is the dominant wave period.

**Theoretical expression of a nonbreaking wave:**

The parameter schemes of vertical diffusion coefficient $K_h$ and vertical eddy viscosity coefficient $K_m$ are

$$K_m = \frac{2ak^2\lambda}{\pi T} e^{\frac{2\pi z}{\lambda}}, \tag{A8}$$

$$K_h = \frac{2Pk^2}{g}\delta\beta^3 W^3 e^{\frac{gz}{\beta^2 W^2}}, \tag{A9}$$

where k is the Kaman constant; a is the amplitude, which is twice the effective significant wave height; T is the fluctuation period; λ is the wavelength; z is the distance from the sea surface to a certain depth; $\beta_*$ is the wave age; P is a dimensionless variable related to the Richardson number coefficient, and δ is wave steepness. Generally, at the sea surface k = 0.4, β = 1.0, P = 0.1, δ = 0.1, π = 3.14, and g = 9.8 m/s$^2$.

**Theoretical expression of radiation stress:**

The components of radiation stress $S_{xx}$, $S_{yy}$, and $S_{xy}$ are calculated as follows:

$$S_{xx} = kE\left(\frac{k_x k_x}{k_x^2+k_y^2}F_{CS}F_{CC} - F_{SC}F_{SS}\right) + E_D, \tag{A10}$$

$$S_{yy} = kE\left(\frac{k_y k_y}{k_x^2 + k_y^2} F_{CS} F_{CC} - F_{SC} F_{SS}\right) + E_D, \tag{A11}$$

$$S_{xy} = S_{yx} = \sqrt{k_x^2 + k_y^2} E \frac{k_x k_y}{k^2} F_{CS} F_{CC}, \tag{A12}$$

where

$$F_{SC} = \frac{\sin hk(z+h)}{\cos hkD}, F_{CC} = \frac{\cos hk(z+h)}{\cos hkD}, \tag{A13}$$

$$F_{SS} = \frac{\sin hk(z+h)}{\sin hkD}, F_{CS} = \frac{\cos hk(z+h)}{\sin hkD}, \tag{A14}$$

$$E = \frac{1}{16} \rho_w g H_s^2, \tag{A15}$$

where $\rho_w$ is the seawater density; $H_s$ is the significant wave height; g = 9.8 m/s$^2$; $k_x$ and $k_y$ are the wave numbers in the x and y dimensions, respectively; D = H + η, where h is the seabed topography and η is the sea surface undulation; and $\int_{-h}^{\eta^+} E_D dz = E/2$ when z ≠ η and $E_d$ = 0.

**Theoretical Expression of Stokes drift:**

The Stokes drift of a single-frequency deep-water gravity wave can be expressed as

$$U_s = U_{ss} e^{\frac{8\pi^2 z}{gT^2}} k \tag{A16}$$

$$U_{ss} = \frac{2\pi^3 H_s^2}{gT^3}, \tag{A17}$$

where $U_s$ is the Stokes drift rate on the ocean surface, **k** is the unit wavenumber vector of the fluctuation, $H_s$ is significant wave height, T is mean wave period, g = 9.8 m/s$^2$, and z is the water depth (z = 0 at the sea surface; z > 0 above the water surface).

**Appendix B**

The Stony Brook Parallel Ocean Model (sbPOM) follows the principles of the POM. The advantage of the sbPOM is that computational efficiency is improved due to its use of a parallel computing environment. However, because the σ coordinate system is used in the vertical direction in the sbPOM, *z* coordinates must be converted into σ coordinates. This conversion is performed as follows:

$$\sigma = \frac{z - \eta}{H + \eta}, \tag{A18}$$

where H(x,y) is the bottom terrain in the horizontal x and y dimensions; η(x,y,t) is the sea level fluctuation in horizontal dimensions x and y at time dimension t, which is integrated from the bottom (z = −H) to the sea surface (z = η); and σ is set from −1 to 0. Using this equation, the basic equations of the sbPOM under the σ-coordinate system can be expressed as

$$\frac{\partial uD}{\partial t} + \frac{\partial u^2 D}{\partial x} + \frac{\partial uvD}{\partial y} + \frac{\partial u\omega}{\partial \sigma} - fvD + gD\frac{\partial \eta}{\partial x} + \frac{gD^2}{\rho_0} \int_\sigma^0 \left[\frac{\partial \rho}{\partial x} - \frac{\sigma}{D}\frac{\partial D}{\partial x}\frac{\partial \rho}{\partial \sigma}\right] d\sigma = \frac{\partial}{\partial \sigma}\left[\frac{K_M}{D}\frac{\partial u}{\partial \sigma}\right] + F_x \tag{A19}$$

$$\frac{\partial vD}{\partial t} + \frac{\partial uvD}{\partial x} + \frac{\partial v^2 D}{\partial y} + \frac{\partial v\omega}{\partial \sigma} + fuD + gD\frac{\partial \eta}{\partial y} + \frac{gD^2}{\rho_0} \int_\sigma^0 \left[\frac{\partial \rho}{\partial y} - \frac{\sigma}{D}\frac{\partial D}{\partial y}\frac{\partial \rho}{\partial \sigma}\right] d\sigma = \frac{\partial}{\partial \sigma}\left[\frac{K_M}{D}\frac{\partial v}{\partial \sigma}\right] + F_y \tag{A20}$$

where

$$D = H + \eta \tag{A21}$$

and where u, v, and ω represent the velocities under the respective σ-coordinates; ρ is the mean fluctuation value; g is the gravitational acceleration; $K_M$ is the vertical viscosity coefficient; and $F_x$ and $F_y$ are the horizontal viscosity terms. The use of these equations

should reduce the truncation errors associated with the calculation of the pressure gradient term in an σ-coordinate system over steep topography.

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
