# Peer review of "A Study of Wave-Induced Effects on Sea Surface Temperature Simulations during Typhoon Events"

_jmse, doi:10.3390/jmse9060622_

Round 1

Reviewer 1 Report

The manuscript entitled “Typhoon-induced Sea Surface Temperature Cooling for Binary Typhoons Events” by Sun et al. represents a significant contribution to the overall (horizontal and vertical) SST cooling effect during binary Typhoon events. I found this manuscript potentially important especially for numerical modelling reasons, and therefore considered its topic as interesting and possibly of great interest for the Journal of Marine Science and Engineering. Overall, all data are sufficient, and the adopted methods are appropriated, as well as the treatment of the data. However, I would like to see the comparison with other referred to the literature events in the same study area or beyond it, on a global scale! I encourage the authors to include such an idea in their revision and make comparisons in the Discussion section accordingly. The figures are appropriated as both quantity and quality. The length of the paper is appropriated for this journal, with all interpretations and conclusions to be in general well justified. Nevertheless, the text is not so well organized, since there are several parts where Results and Material and Methods have mixed each other, and this makes the manuscript not easily readable and understandable. The Introduction should be rewritten at some places and make it to be to the point, highlighting the main objectives of this work. Finally, the bibliography is accurate and updated. Although I am not mother language, I could read the manuscript easily. The English is in relatively good shape, but several places need significant improvements. However, there are some critical points that need to be clarified and/or better discussed before acceptance (see my comments below). Therefore, I propose a few points to be addressed before it can be considered for publication (minor revision) in Journal of Marine Science and Engineering. Both the suggestions and minor comments are listed right below. It is accepted with ninor revision.

Major issues/minor comments and suggestions:

-L34: “It is well known” Rephrase it…Try to start the introduction with another way

-L43: …have been studied…

-L42-54: This paragraph seems like a background info-related paragraph for the methods applied. Actually, it is out of space. I propose to the authors to create a new sub-section after the Introduction and include there all this information entitled “Development or Advances of methods” or something like that

-L86: Which are these effects? Mention them briefly

-L90: Delete or rephrase this sentence. Possibly, it is redundant

-At the end of the Introduction, a paragraph explaining the main goal of this work is missing and certainly should be added. As it is, the manuscript lacks focus, and it is not attractive at all for the reader to go on reading it

-L113-118: These are results and not material & methods. So, they should be moved at the Results section

-L129-131: Again, the same. This is out of space. Eliminate it from here

-L159-161: These introductory sentences are not needed. It is not a book chapter to describe what will follow, but a scientific article

-L190-191: I encourage the authors to add some more information related to the differential path-related effect on SST cooling, explaining it in detail. Which are their characteristics? When do they formed, under what regimes etc?

General comment: In the Discussion, what is missing, is the reference to other related events from the literature in the same study area or beyond, and a more thorough comparison of SST differences, depths, pathways etc

-L225: Delete the double-referred e.g.,

Author Response

The replies are in the attached document.

Reviewer 2 Report

The manuscripts presents how including wave related processes in an ocean model can increase the cooling of the sea surface during typhoons. Both the wave model results and sea surface temperatures are validated using available in-situ and remote sensing data.

The study itself seems well done and the topic fits the scope of the journal. I therefore generally have a positive attitude to the study, since the effect of waves in hydrodynamical simulations are definitely a worthwhile subject to study. Although the study is interesting, I found the actual manuscript a bit lacking. The major flaws are that the language is (at times) bad, the materials and methods are lacking in clarity, and the presentation of the results are very brief.

I therefore recommend that this manuscript should undergo major revisions, but feel confident that the study in itself is interesting and well enough executed that it can be published after the flaws in the manuscript has been corrected. I will now expand on my opinion of the major flaws:

1) Bad language

I don’t hold imperfect English against the authors, but now the language hinders the understanding of what is trying to be communicated. The text also contains several errors that could have easily been corrected (although I don’t think these are a major issue).

Examples of the unclear language are:

Lines 108-110: “Taking advantage of the ECMWF wind data, the long-term wave distributions simulated from SWAN [39] and WW3 [40] models are well analyzed.”

What does “well analyzed” mean? That the results are good or that it the product covers large scales, or that the analysis is being done well? Who is analysing and for what region? Reading the references it is clear that they are to wave hindcasts, but how these hindcasts relate to this study is not clear at least from the text. Please be a bit more specific.

Line 113: “It is found”

Who found? Is this done in this study? We are in the materials and methods, so I’m not expecting results here, but no other study is cited.

Line 210: “effect of SST cooling gradually becomes weak”

What does this mean? How is the cooling of the sea surface temperature effecting something, and what? Or are you talking about the effect of the wave related processed on the water temperature?

Lines 230-231: “This kind of winds could improve [...]”

Didn’t you already use these wind in this study? Now it sound like using these winds are recommended as further research.

Lines 237-238: “[…] the accuracy o sbPOM simulated SST is improved by about 0.5 oC”

Models don’t have a general “accuracy”. Are you talking about bias, etc.?

Title: I still don’t know what a “Binary” Typhoon is and how it differs from a normal Typhoon and why we should study binary typhoons in particular.

Smaller examples:

Line 225: “e.g., e.g.”

Line 276: “mdoe” and “descried”

2) Lacking clarity in material and methods

WAVEWATCH is described in the appendix, but no description of the source terms are provided (WW3 has meny options). Also, equations A3-A6 are not really explained, since delta_1 and W, and gamma etc. are not defined. It is also unclear why you are parameterizing the wave breaking when WW3 can output the wave breaking directly. Also, why are you using a drag coefficient to calculate the friction velocity when WW3 has an iterative approach that accounts for the wave stress when determining u*.

The coupling of the wave processes to the ocean model is not explained. I understand that wave breaking enhances mixing, but how is this number that comes from the wave model connected to the ocean model. Is the Stokes drift simply added on top of the surface current in the model, or is the decay of the Stokes drift with depth accounted for? Is the interplay between the friction velocity and Stokes drift that can cause Langmuir circulations accounted for etc. You need to provide more specifics here. The wave model field are not even listed as Forcing fields in Table 1.

Didn’t the ECMWF data contain any radiation etc. data? Now you are using ECMWF winds and NCEP radiation and heat flux forcing. Is there a reason for using to different data sets? Do you expect that this affects the results?

3) Brief results section

The results section doesn’t have to be overly long, but now most of the results are just model validations (although I welcome the use of measurement data for the validation, so definitely don’t remove this). However, the title of the paper promised more than a model validation, and the results that the title promised are in the Discussion part, although also there quite briefly. Since the discussion is just new results, then it also means that there is no read discussion.

Lastly we come to what I think I the main flaw of the paper. The effects of the wave related processes are not evaluated separately, so we get no grasp on which one is important and which one is not. This deficiency, together with the rather poorly documented coupling methodology, severely limits how useful this paper could be to the scientific community.

Author Response

(The authors gave the same response as above.)

Reviewer 3 Report

The paper discusses the cooling pattern induced by two sets of almost simultaneous typhoons (so-called “binary typhoons’) in the China Seas. The authors used the simulated wave data from WWIII incorporated in the sbPOM circulation model to compare the effect of waves on SST cooling during each pair of typhoons. They evaluated the wave model with satellite data and showed that including four different components of the wave-induced hydrodynamics/mixing would improve the results for SST during the typhoons.

Although the topic is interesting and is within the category of the JMSE topics, the paper outline and title are inconsistent and the authors substantially failed to establish an appropriate scientific story and just showed some simulation results. Therefore I cannot recommend this paper to be published in the JMSE. I hope that given the fact that the topic is interesting, this emission does not discourage the authors and try to re-submit the paper with improved quality.  Followings are more details about the reasons that I made this decision:

  • The paper title is not consistent with the paper content. In the title, the main keywords and emphasis are “ Typhoon-induced cooling’ and ‘binary typhoons’, while in the paper great amount of works were done to describe and validate the WWIII wave model, and eventually, it was the effect of waves that were compared to the non-wave condition.

  • Almost nothing was mentioned about the effect of ‘binary typhoon’ on the cooling pattern and the contribution of each typhoon in producing the resulted cooling pattern. You cannot just say that the pattern is complex and make no attempts to clarify and analyze the pattern. Specifically, one needs to know that what is specific about binary typhoons. How the simultaneous effect of two typhoons changes the cooling pattern compared to just one typhoon? The authors need to address these issues by noticing the cooling effect of a single typhoons like some classic studies by Price (1981) and Bender et al(1993).

  • The contribution of the four wave-induced mechanisms in changing the mixing depth and SST are not the same. In fact, some of them like the effect of the radiation stress can be ignored and the effect of non-breaking waves could be essential (see Ajaiz et al., 2016; Nonbreaking wave-induced mixing in upper ocean during tropical cyclones using coupled hurricane-ocean-wave modeling).

  • Not much about the model results except some maps of SST or vertical profiles at locations is presented and discussed. One expects that the mixing during the ‘binary typhoons’ is discussed given a complete set of available model results.

  • The English language of the paper is really difficult to follow. Most parts of the paper have many sentence structures and grammatical errors.

Author Response

(The authors gave the same response as above.)

Round 2

Reviewer 2 Report

This is the second time I am reviewing this manuscript. The first time I recommended major revisions. I am pleased to see that the authors have made many significant improvements to the manuscript. Especially, my main concern about the effect of the different wave-induced effect not being evaluated separately has now been addressed (Table 2). The language is also much improved and the manuscript is now easy to follow.

I still have a few comment that will probably be quite easy to fix. I can therefore recommend that this manuscript is accepted for publication after minor revisions.

I have two main comments and then also list some technical corrections/suggestions:

Main comment #1:

The structure of the manuscript is still a bit weird, since a lot of the main results (about the profiles etc) is in the discussion. One possibility might be that the authors create a new subsection for the results they now have in the discussion, and then simply end with a “Summary and conclusions” instead of “conclusions”. The discussion would then be assimilated into the results section. I usually prefer a separate discussion, but I think the above solution might work in this case (as long as it is in accordance with journal guidelines and ok'd by the editor).

The other possibility is to move the results out of the discussion and then expand the discussion section with new material so that it stands on its own. I will leave it up to the authors and editor to decide how to proceed, but the results need to be moved from the discussion section to the results section.

Main comment #2:

I think the wave model description is still a bit lacking. This seems like minor things to go on about, but I think they are important to ensure replicability of the results (and to give other researchers credit for their work). I will point out specific flaws:

a) The source term package is only mentioned by name without citations. I think ST2 is Tolman & Chalikov. Developing a new source term package is no small feat, so I think we should cite the original authors and give the the credit they deserve. I’m sure the appropriate citation can be easily found in the WW3 manual.

I also think that DIA should reference the work of Hasselmann. Also the other source terms (Sbot, Sdb) has different options in WW3 (JONSWAP, SHOWEX etc.), so please specify these also.

b) No reference is provided for Equation A4.

Eq. A6 doesn’t make sense, since the model doesn’t model H1/3, but Hm0 (they are approximately the same in deep water, but A6 is not the definition for H1/3). This is just a notation issue if you, indeed, used Hm0 (as you should).

Equation A7: I would think that the model outputs the peak period (Tp) directly. We could then define the angular peak frequency directly as wp=2*pi/Tp, and this is how it should be done if at all possible. T1/3 is also not properly defined, since it is only called the “wave period”. It looks like it is the mean wave period of the highest one third of the waves, but this is not something the model calculates. If you didn’t for some reason have Tp, then specify what period you are using (Tm01, Tm-10 etc) and cite a reference for the coefficient (here 0.91) you are using.

Provide a reference for what parameterization of the drag coefficient (Cd) you are using.

E, as defined in Eq. A6, is not the spectral density, but the variance of the wave field. E(f) (given as a function of frequency) is the spectral density (or variance density), but the “density” part vanished once we integrate.

List of technical corrections:

#1) Line 15: “third-generation”, not “three-generation”. I also think WAVEWATCH III is written without the dash (-).

#2) Line 50: “XX”. Perhaps you meant to cite the work of the WAM group?

#3) Lines 56: WW3 can use unstructured grids and SWAN can use structured grids, so this description of the models is incorrect. Perhaps you can say that SWAN was originally developed as a nearshore model, while WW3 was developed for the oceanic scales, but the current version contains the necessary options to use it also for coastal applications? (I don’t think you need to explain the differences in too detail, nor do I think you somehow need to defend your use of WW3).

#4) Line 84: Missing a full stop. “[…] winds. Thus, [...]”

#5) Line 91: “processors”. Do you mean predecessors?

#6) Line 112: “constructed”. This sound like you built new models. Perhaps “implemented”?

#7) Table 1 still doesn’t mention the wave-induced processes in the “Forcing fields” of sbPOM (I already commented on this in my previous review). If the authors have a good reason to leave them out, then ok, but I think it is a bit confusing for the reader.

#8) Figure 9 and 10. I think the start of the figure captions (and the colorbar labels) should reflect that these are differences in SST, not SST. So colorbar could read “Difference in SST from sbPOM (oC)” (you have already defined SST). And the captions could start “Difference in daily average SSTs [...]”. I think it would make it clearer.

#9) Isn’t figure 12 just the vertical temperature profile? The SST is defined as being at the surface, so it can’t have a profile going down to 500 metres.

Author Response

We truly thank the anonymous reviewers for the constructive comments that have helped us to improve this manuscript. We provide a point-by-point reply in the document to carefully address these comments and suggestions.

Reviewer 3 Report

I appreciate that the authors addressed most of the issues in the paper. They need to revise their response to comment2  as follows to wrap up their revision:

About the response to Comment 2:

The authors need to do more than just citing the papers suggested by the reviewer to compare the response under a ‘single typhoon’ and ‘binary typhoons’. For a single typhoon (hurricane), the authors can add something like the following statement to the introduction:

 “The produced cooling is a function of both typhoon forward speed and intensity. Generally, the lower is the forward speed, the higher is the cooling and the higher is the intensity, a larger cooling is expected (Allahdadi., 2014; Allahdadi et al., 2017a). The extra cooling and turbulent mixing on the right side of the track in the northern hemisphere as a result of the rightward bias can contribute to a larger deepening of the mixed layer (Allahdadi et al., 2017b).”  (See the citations below)

 It would be appropriate if the authors can add a statement about how the probable response of the ‘binary typhoons’ compares to the above response from a single typhoon.

Allahdadi, M., 2014. Numerical Experiments of Hurricane Impact on Vertical Mixing and De-Stratification of the Louisiana Shelf Waters. LSU Doctoral Dissertations. 

Allahdadi, M.N., Li, C., 2017a. Numerical Simulation of Louisiana Shelf Circulation under Hurricane Katrina. Journal of Coastal Research. https://doi.org/10.2112/JCOASTRES-D-16-00129.1

Allahdadi, M.N., Li, C., 2017b. Effect of stratification on current hydrodynamics over Louisiana shelf during Hurricane Katrina. Water Science and Engineering 10, 154–165. https://doi.org/10.1016/j.wse.2017.03.012

Author Response

(The authors gave the same response as above.)
